# The Presence of the Biochar Interlayer Effectively Inhibits Soil Water Evaporation and Salt Migration to the Soil Surface

**Qiang Xu** [1,2], **Hongguang Liu** [1,2] , **Mingsi Li** [1,2,*] and **Pengfei Li** [1,2]

1. College of Water Conservancy and Architectural Engineering, Shihezi University, Shihezi 832000, China
2. Key Laboratory of Modern Water-Saving Irrigation of Xinjiang Production and Construction Group, Shihezi 832000, China
* Correspondence: limingsi@shzu.edu.cn

**Abstract:** To reveal the mechanisms of water conservation and soil salinity control in the biochar interlayer, the effects of biochar addition as an interlayer on soil water infiltration, evaporation, and salt transport were studied. Through the indoor soil-column simulation test, soil columns were set up by packing homogeneous soil (CK) and biochar spacers into columns at different burial depths of 10, 20, and 30 cm. The biochar interlayer decreased the infiltration capacity of the soil, with the average infiltration rate decreasing from 0.72 cm·h$^{-1}$ to the ranges of 0.39–0.48 cm·h$^{-1}$ in the CK soil column, and salt leaching efficiency was improved. The salt content in the bottom layer of soil in the CK column was reduced to within the range of 19.96–47.46% compared with that in the barrier soil column. The presence of the biochar interlayer improved the distribution of soil water and salt. The soil water content in the upper layer above the interlayer was around 7.79–13.68% higher than that in CK, whereas the average salt content was 6.44–60.40% lower than that in CK. The biochar interlayer inhibited soil water evaporation, and cumulative evaporation in this layer decreased by 32.34–42.10% compared with that in CK. The salt accumulation in the interlayer in the soil column decreased within the range of 16.36–51.36% compared with that in the CK soil column. The biochar interlayer could not only retain water for a long time, but also adsorb the salt leached from the upper layer, and thus, inhibit the reverse salt flux from the lower layer. The creation of the biochar interlayer of 30 cm could play a role in soil salinity control and water conservation, and can also provide a basis and reference for the improvement of saline-alkali farmland in arid and semi-arid areas.

**Keywords:** biochar; infiltration; evaporation; interlayer; water salt migration





## 1. Introduction

Xinjiang is the largest saline-alkali land area in China, with a total area of $2.20 \times 10^7$ hm$^2$ [1], of which 31.10% of the existing farmland is affected by soil salinization [2]. Soil salinization and a shortage of freshwater resources are the direct factors limiting farmland use efficiency and causing low farmland productivity [3]. Due to special climate conditions in Xinjiang, there are many problems facing the saline-alkali farmland, such as a high evaporation drop ratio, shallow groundwater level, and salt accumulation [4,5]. Therefore, reducing soil water evaporation and inhibiting salt migration to the soil surface have both become the most effective ways to improve the saline-alkali soil [6,7]. Plastic-film mulching can reduce water evaporation from the soil surface, conserve water, control salt accumulation, improve soil water and heat processes, activate soil nutrients, and increase crop yield in Xinjiang [8–12]. However, long-term plastic film mulching on a large scale has caused the accumulation of plastic film residuals in farmland. The average plastic film residue in Xinjiang exceeded 265.3 kg·hm$^{-2}$, 4.5 times the national average [13]. Zhang et al. [14] believed that straw mulching could be an effective means of reducing plastic film pollution in farmland soils, and it has gradually attracted attention in recent years. Zhao et al. reported that, at a depth of 20–40 cm, the underground straw cover could inhibit the

upward migration of water and salt to the deep layer. The soil salt content is low, and salt mainly accumulates in the deep layer, which is conducive to the desalination of water in the root zone of crops. The salt inhibition effect is stronger in this layer than in the surface layer and can play a role in the inhibition of salt accumulation and improvement of saline-alkali soil in the interlayer [15–18]. However, deep straw mulching requires high levels of inputs such as labor and machine, with a short-term effect (about 100 days), and has no obvious inhibition of evaporation and salinity in the late stages of the growth of crops [6,18], which limit its large-scale promotion and application. The preparation of biochar from a straw can considerably increase the duration of effectiveness of deep straw mulching. Biochar has characteristics such as high carbon content and stable physical and chemical properties [19], which can remain unchanged in soil for hundreds of years [20]. Moreover, biochar has been widely used as a soil conditioner [20,21] to improve soil water retention [22], reduce soil salinity and alkalinity [23], increase soil CEC and nutrient and optimum water content [24], improve crop nutrition [25], and enhance crop yield [20,26]. Most previous studies on biochar have revealed its addition to the soil surface, while studies on the processes of soil water infiltration and evaporation and the optimization of soil water and salt distribution using biochar as a barrier have rarely been reported. In this paper, the indoor soil column test was used to study the effects of the biochar interlayer on soil water infiltration and soil water evaporation, and unravel its regulation mechanisms of soil water and salt movement to provide a theoretical basis and reference for water conservation and salt removal in the saline farmland in arid and semi-arid areas.

## 2. Materials and Methods

### 2.1. Test Materials

The test was conducted in the Water Conservancy and Civil Engineering Experiment Center (86° 03′ E, 44° 18′ N, 451 m above sea level), the School of Water Conservancy and Building Engineering, Shihezi University, from April 2020 to December 2021. The soil for the test was taken from the experimental farm of Shihezi University, and the sampling depth was 20–40 cm. Using the hydrometer method, the mass fraction values of each particle group in the soil were 14.86% sand (0.02–2 mm), 80.70% silt (0.002–0.02 mm), and 4.44% clay (0–0.002 mm). The soil texture was silty loam. According to the soil salt content, saline soil is classified as soil with chloride and sulfate salts. Before conducting the test, the soil transferred to the laboratory was dried, ground, and passed through a 2-mm sieve after removing impurities. Soil was mixed with the prepared NaCl and $Na_2SO_4$ mixed aqueous solution with a mass ratio of 1:1. Due to the availability of biomass carbon in the study area, cotton straw-derived carbon was selected as the test biomass carbon. The cotton stalk was chopped into small segments of 10–20 cm by a straw chopper and pyrolyzed for 3.5 h at about 350 °C under incomplete anoxic conditions (see Table 1 for the physical and chemical properties of soil and biochar).

**Table 1.** Physical and chemical properties of soil and biochar.

| Texture | Soil Bulk Density /g·cm$^{-3}$ | Field Water Holding Capacity/% | Salt Content /g·kg$^{-1}$ | Organic Carbon /g·kg$^{-1}$ | Total N /g·kg$^{-1}$ | Total P /g·kg$^{-1}$ | K /g·kg$^{-1}$ | Ca /g·kg$^{-1}$ | Mg /g·kg$^{-1}$ |
|---|---|---|---|---|---|---|---|---|---|
| Silty loam | 1.40 | 26.46 | 26.56 | 4.63 | 0.39 | 0.75 | 12.35 | 6.28 | 4.51 |
| Biochar | 0.51 | — | 3.36 | 521.69 | 25.72 | 11.02 | 20.58 | 19.63 | 4.26 |

### 2.2. Test Design

The test treatments were divided into four groups, namely CK, H1, H2, and H3. The carbonized cotton straw chopped to a length of about 5–10 cm was evenly placed at distances of 10 cm, 20 cm, and 30 cm away from the soil surface so that a biochar spacer

was laid, and then 200 g of carbonized straw was added to each soil column, with the thickness of the straw spacer of 5 cm after compaction. The total depth of the homogeneous soil layer, along with the gravel-sand layer and the straw interlayer in the final soil column, was 70 cm. The soil column used was a cylindrical organic glass column with a height of 100 cm and a diameter of 20 cm. The bottom of the soil column was sealed, and an exhaust hole was created in the middle. Three sampling holes were evenly created at an interval of 5 cm around the soil column from its bottom to the top. During the test, the sampling holes were blocked with rubber plugs. The Markov bottle was used as a constant water supply device, with a cross-sectional area of 64 cm$^2$ and a height of 80 cm. The pressure water head of 1.5 cm was set during the water supply. Before loading soil, the 5-cm thick sand and gravel were laid at the bottom of the soil column as the inverted filter, which could provide a smooth air infiltration for facilitating the infiltration process. To prevent the movement and entrance of the fine-grained soil from the upper layer to the large pores in the sand layer, two layers of nylon cloth with the same cross-sectional size as the soil column were laid on the sand layer. The soil samples were loaded into the soil column in equal quantities at the set mass per unit volume of 1.40 g·cm$^{-3}$. To make the consistent mass per unit volume of the soil column, the thickness of the soil layer laid each time was controlled and set to 5 cm. After the compaction of one layer, its surface was brushed, and then the next layer was placed to ensure that the compacted soil layer was uniform and gave good plant root-soil contact (Figure 1 for test device).

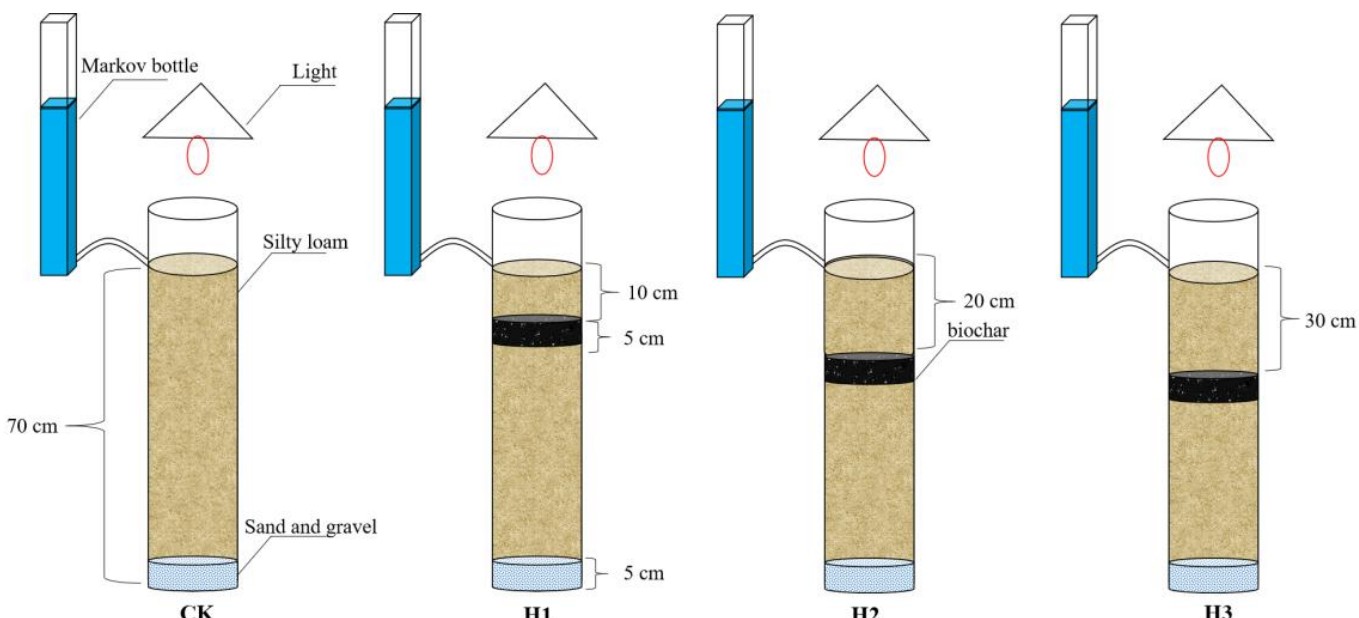

**Figure 1.** Schematic diagram of the test device.

### 2.3. Test Operation Process

The soil water infiltration test device is composed of a soil column, water supply system, and iron frame platform. After the soil column was filled, a one-dimensional vertical soil column ponded infiltration test was conducted, using deionized water. Water was added to each soil column to keep the depth of the water layer at 1.5 cm, which was controlled by a Markov bottle to remain unchanged throughout the test. During the test, the position of the wetting front and the water level in the Markov bottle at different infiltration times were recorded to obtain the values of cumulative infiltration amount corresponding to different infiltration times. When the wetting front of each soil column reached 50 cm, the water supply was stopped, and the pipe orifice of the soil column was immediately sealed with plastic film to prevent the natural evaporation of moisture from the soil surface. The irrigation amounts of the CK, H1, H2, and H3 soil columns were 97.05 cm, 96.22 cm,

95.05 cm, and 95.41 cm, respectively. After the completion of all water infiltration, the soil water infiltration process was completed.

### 2.4. Soil Sample Collection and Determination Method

(1) Soil water and salt content determination: before conducting the soil water evaporation test, each soil column was sampled in different soil layers through the sampling hole as the initial condition of the evaporation test. Sampling was performed at a depth of 0–60 cm, with a sampling interval of 5 cm for each sample. Soil samples were taken at 15, 25, and 35 days after evaporation, and the sampling depth was consistent with that before evaporation. The sampling holes were created at the same height. The sampling positions were not the same at each sampling time. After removing the soil, the same-moisture-content parent soil was used for backfilling. The soil moisture content was determined after sampling using the drying method, and the soil electrical conductivity was determined using a leaching solution (1:5 soil water ratio). The fitting formulas of soil electrical conductivity and soil salt content are as follows:

$$S = EC_e \times 4.30 + 0.16 \ (EC \leq 1 \ ms \cdot cm^{-1}) \tag{1}$$

$$S = EC_e \times 6.75 - 1.90 \ (EC > 1 \ ms \cdot cm^{-1}) \tag{2}$$

(2) Soil strength and cumulative soil water evaporation: During the test, on every sampling day at 10:00 a.m., an electronic balance was used to weigh each soil column, and the values were recorded. The evaporation strength is the difference between the weights of the soils on two adjacent days, and the cumulative evaporation is the difference between the weights of soils on the first day and the 35th day (Zhao et al.) [7].

(3) Water and salt content of the biochar interlayer: soil samples were taken at 0, 15, 25, and 35 days after evaporation. After sampling, soil moisture content was determined by the drying method, and soil electrical conductivity was determined using the leaching solution (soil water ratio of 1:5). The fitting formulas of soil electrical conductivity and soil salt content are the same as mentioned above.

(4) The Kostiakov infiltration formula is: $I(t) = kt^n$, in which k is the cumulative infiltration in the first unit of time (mm) and n is the empirical constant from the fit of experimental results to the equation (Cen et al.) [27].

(5) The relationship between cumulative evaporation and time was fitted by the Rose empirical formula, as follows: $y(t) = \omega t + \beta T^{0.5}$; where $y(t)$ is the cumulative evaporation (mm); $\omega$ is the stable evaporation parameter; and $\beta$ is the water diffusion parameter. Please refer to the research results reported by Zhan et al. [28].

### 2.5. Data Processing

The data used in this experiment were the average values of three replicates. The test data were analyzed by Excel 2003 (Microsoft, Redmond, WA, USA), and data fitting and statistical analysis were performed by the SPSS 19.0 (IBM, Armonk, NY, USA).The average values of the parameters in each treatment were calculated, and the correlation analysis was conducted. The LSD method was used for calculating the significance between the means of different treatments ($p < 0.05$), and Origin 2020 (OriginLab, Northampton, MA, USA) was used for mapping.

## 3. Results and Analysis

### 3.1. Effect of the Biochar Interlayer on Infiltration Characteristics

The infiltration depth of the wetting front of each soil column increased during the infiltration process, and the presence of the biochar interlayer significantly reduced the advancing velocity of the wetting front, making the infiltration process in this layer different from that in the CK soil column (Figure 2). The wetting front of the CK soil column uniformly moved during infiltration. When the wetting front of the H1, H2, and H3 soil columns moved through the soil layer above the interlayer, the migration process was the

same as that of the CK soil column. Once the wetting front entered the biochar interlayer, its moving speed rapidly decreased, and therefore, it took significantly longer to pass through the interlayer than the time taken for the wetting front in the CK soil column to pass through the same soil layer. The presence of the straw interlayer reduced the advancing velocity of the wetting front, which induced impermeability. The time taken for the treatments, including CK, H1, H2, and H3, to reach the depth of 50 cm was 70 h, 95 h, 105 h, and 115 h, respectively, and the average infiltration rate decreased from 0.72 cm·h$^{-1}$ in the CK soil column to 0.52 cm·h$^{-1}$ (H1), 0.48 cm·h$^{-1}$ (H2), and 0.43 cm·h$^{-1}$ (H3).

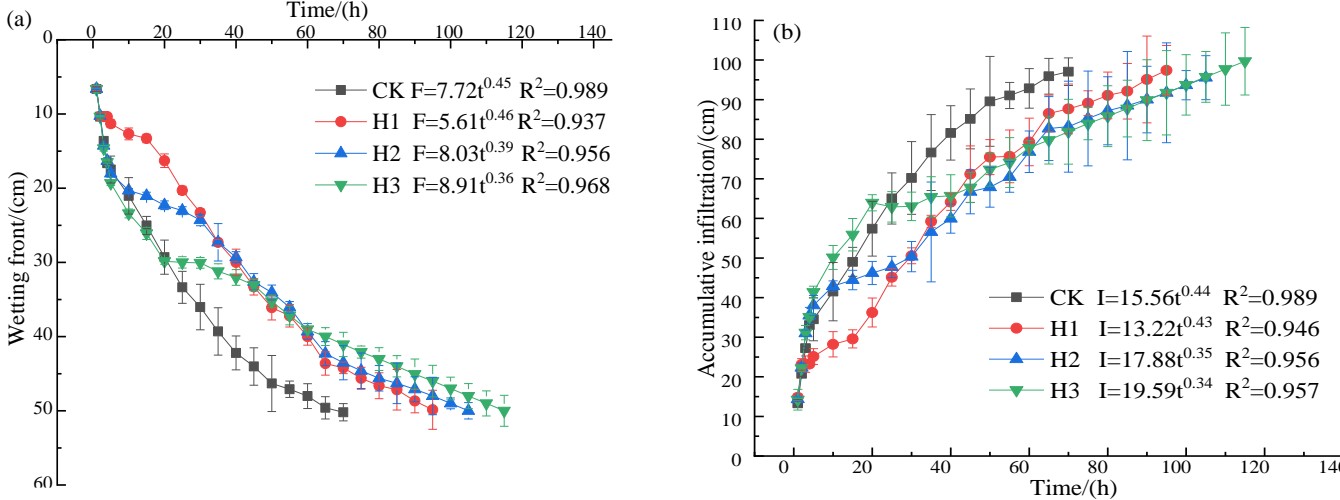

**Figure 2.** Effect of the biochar interlayer on infiltration characteristics. (**a**,**b**) represent the change processes of the wetting front depth and cumulative infiltration, respectively, with time for each treatment.

The cumulative infiltration amount of each soil column increased with time, and the presence of the biochar interlayer significantly reduced the water infiltration rate. When the wetting front moved through the soil layer above the interlayer, the cumulative infiltration amount of each soil column showed the same trend with time. When the infiltration water entered the interlayer, the cumulative infiltration amount of H1, H2, and H3 soil columns per unit of time rapidly decreased, 37.60%, 63.02%, and 83.85% lower than that of the CK soil column, respectively.

### 3.2. Effect of the Biochar Interlayer on Soil Profile Water Content

It can be seen from Figure 3 that the soil water content in each treatment gradually decreased with the deepening of the soil layer, and the decreasing trend in the biochar interlayer treatment was more prominent. Since the influence of different depths of interlayer on soil water infiltration was different, there was a variation in the soil moisture content between soil columns. The average water content of the H1 soil column at the 0–15 cm soil layer was 13.68% higher than that of CK, while the average water content of the 15–60 cm soil layer was 7.74% lower. The average water content of the H2 soil column at the 0–15 cm soil layer was 8.39% higher than that of CK; whereas, in the 15–60 cm soil layer, the average water content was reduced by 10.95%. Before evaporation, the average water content of the H3 soil column at the 0–30 cm soil layer was found to be 7.79% higher than that of the CK soil column. The soil layer of 30–60 cm, however, presented the opposite trend, and its average water content decreased by 17.31% compared with that in CK.

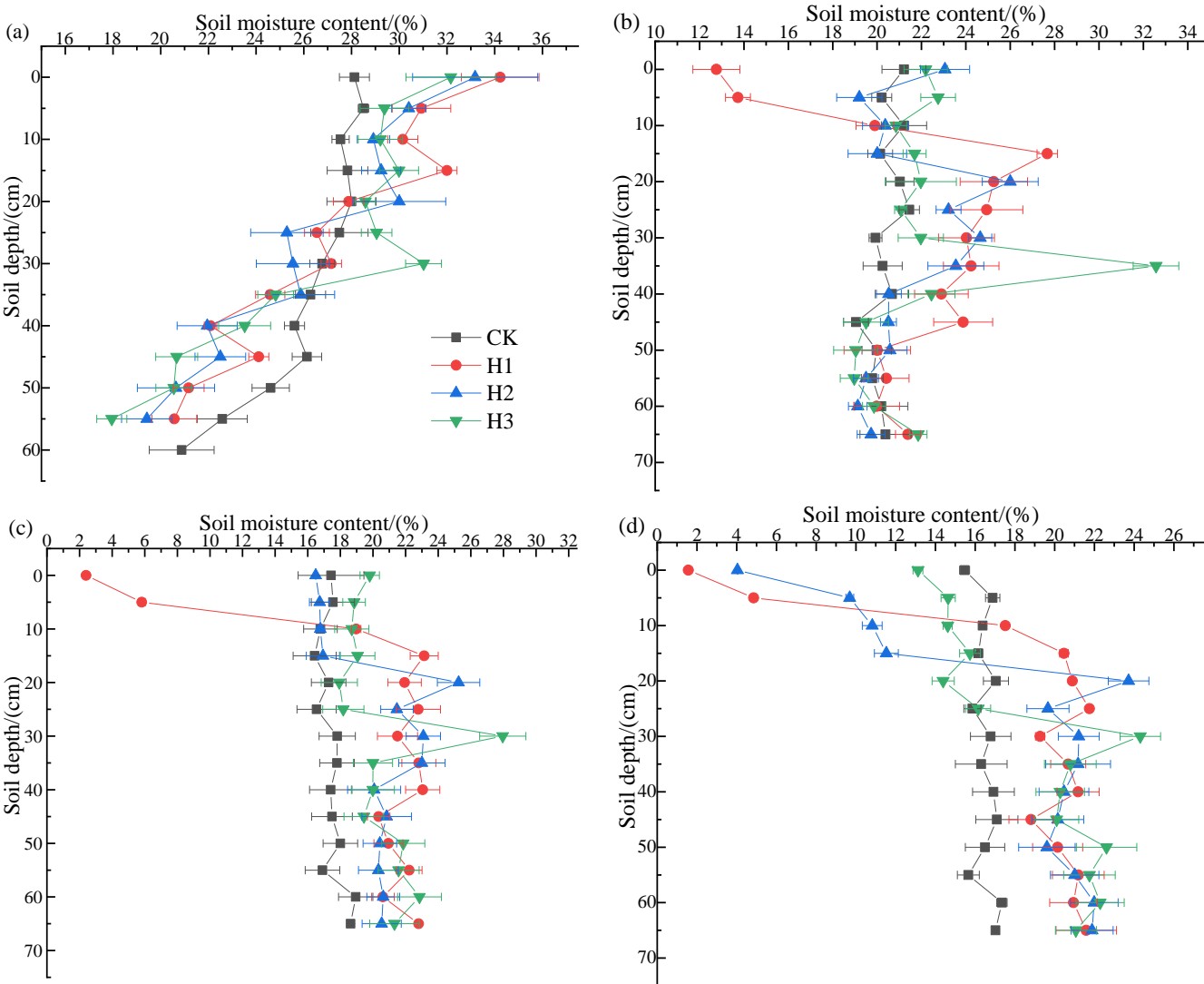

**Figure 3.** Soil water content profiles with evaporation time for different treatments. (**a**–**d**) represent the soil moisture content at 0, 15, 25, and 35 days of evaporation, respectively.

During the evaporation process, the dehydration process in each soil column was different, and the water content in all soil layers of the CK soil column was uniformly reduced, but the soil water content of the whole soil profile was not different. The water content in the soil layer above the interlayer in H1, H2, and H3 soil columns decreased, whereas the water content below the interlayer remained at a high level. Within 35 days of continuous evaporation, the average water content at the 0–10 cm soil layer in the H1 soil column decreased within the ranges of 35.00–103.55% compared with that in CK, while the average water content of the 10–60 cm soil layer increased within the ranges of 14.79–24.80% compared with that in CK. During the first 25 days of evaporation, the average water content in the 0–20 cm soil layer of the H2 soil column was 3.92–6.43% higher than that of the CK soil column, whereas on the 35th day of evaporation, the average water content of the 0–20 cm soil layer was 35.81% lower than that of the CK soil column. The average water content of the 20–60 cm soil layer, however, was 25.54% higher than that of the CK soil column. During the first 25 days of evaporation, the average water content of the 0–30 cm soil layer in the H3 soil column was around 11.60–16.34% higher than that in the CK soil column. On the 35th day of evaporation, the average water content of the 0–30 cm soil layer was 0.65% lower than that of the CK soil column, while the average

water content of the 30–60 cm soil layer was 28.40% higher than that of the CK soil column (Figure 3).

### 3.3. Effect of the Biochar Interlayer on the Salt Content of the Soil Profile

Since the depth of different interlayers affected the distribution of water content of the soil profile, the salt content of the soil profile also significantly changed (Figure 4). The curve of the salt content of the soil profile of the CK soil column was smooth, whereas that of the soil profile salt content of the interlayer soil column fluctuated. Before evaporation, the total salt content of the soil layers above the H1, H2, and H3 soil column barriers decreased by 21.89%, 7.38%, and 26.42%, respectively, compared with that of CK, and the cumulative salt content of the 0–30 cm soil layer decreased by 47.10%, 33.23%, and 26.42%, respectively, compared with that of CK.

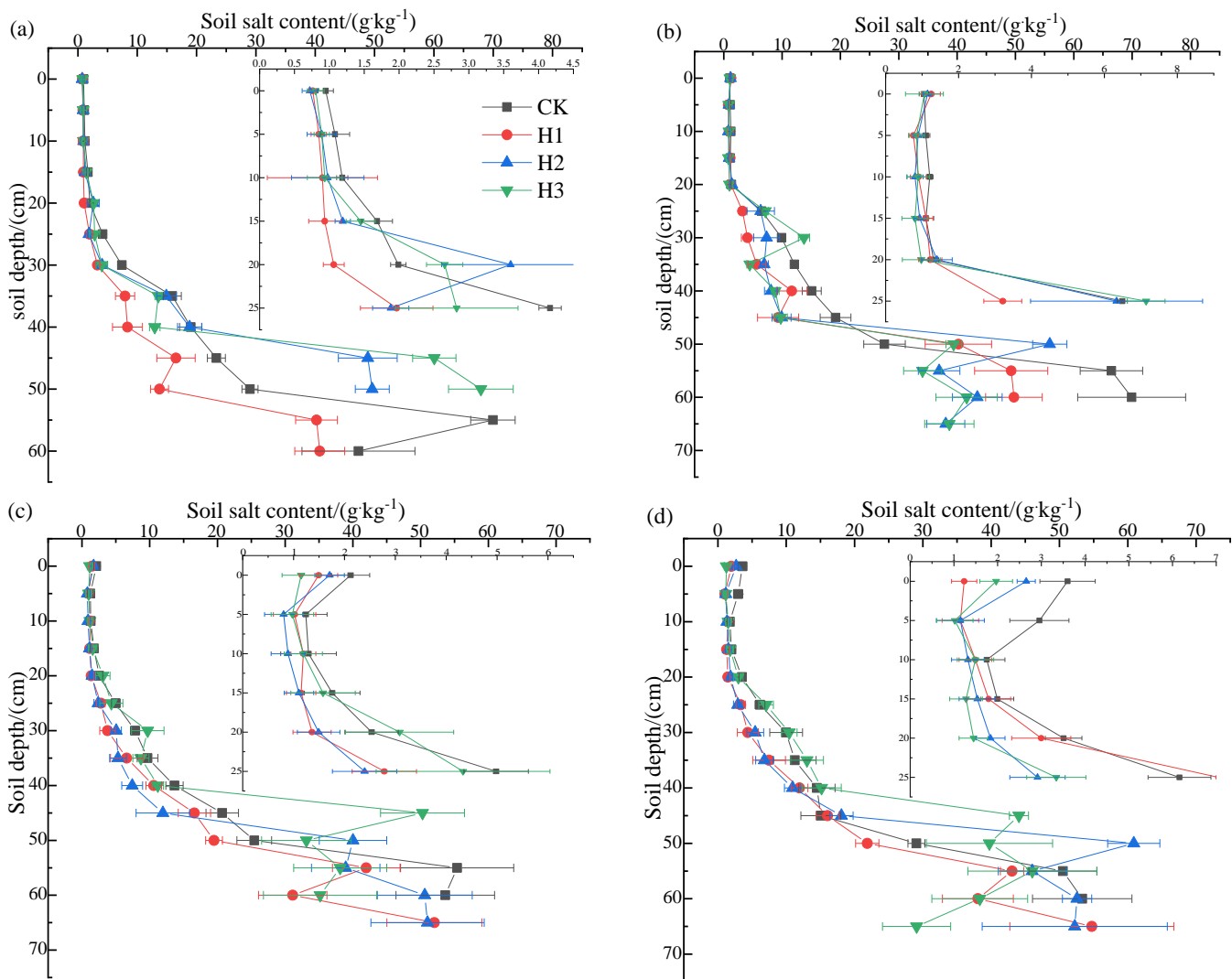

**Figure 4.** Salt content in soil profiles with evaporation time for different treatments; (**a**–**d**) represent the soil salt content at 0, 15, 25, and 35 days of evaporation, respectively.

During evaporation, the salt in the bottom soil layer and phreatic water could move upward with water flow, resulting in salt accumulation on the surface of the soil column. On the 15th day of evaporation, due to the presence of the biochar barrier, salt content in the 0–5 cm of H1, H2, and H3 soil columns was 15.96%, 8.65%, and 1.26% higher than that of the CK soil column, respectively. The soil below the barrier was not affected by evaporation, and the salt continued to move downward. From the 25th day of evaporation,

the salt content of the surface soil of the CK soil column exceeded that of the soil column treated with the biochar interlayer. On the 35th day of evaporation, the salt content of the surface soil for each treatment was the highest, and the salt content of the surface soil of the CK soil column was significantly higher than that of the soil column treated with the biochar interlayer. The total salt contents of the soil layer above the interlayer in H1, H2, and H3 soil columns were 50.17%, 38.46%, and 53.32% lower than that of CK, respectively, and the cumulative salt content of the 0–30 cm soil layer had the values of 50.17%, 43.68%, and 12.64% lower than that of CK, respectively.

### 3.4. Effect of the Biochar Interlayer on Evaporation Intensity and Cumulative Evaporation

Figure 5a shows the influence of the biochar interlayer on the evaporation intensity. The evaporation intensity of the CK and H3 soil columns was relatively high in the first 15 days of evaporation, and then, it gradually stabilized and finally remained at a relatively high rate, while the evaporation intensity of the H1 and H2 soil columns was similar to that of the H3 soil column in the first 15 days of evaporation and then significantly decreased. The biochar interlayer could significantly reduce the evaporation intensity, with a more pronounced effect in the H1 soil column than in the H2 and H3 soil columns. The average evaporation intensity of CK, H1, H2, and H3 soil columns during the 35 days of evaporation had values of 91.19 $g \cdot d^{-1}$, 66.20 $g \cdot d^{-1}$, 69.59 $g \cdot d^{-1}$, and 71.45 $g \cdot d^{-1}$, respectively.

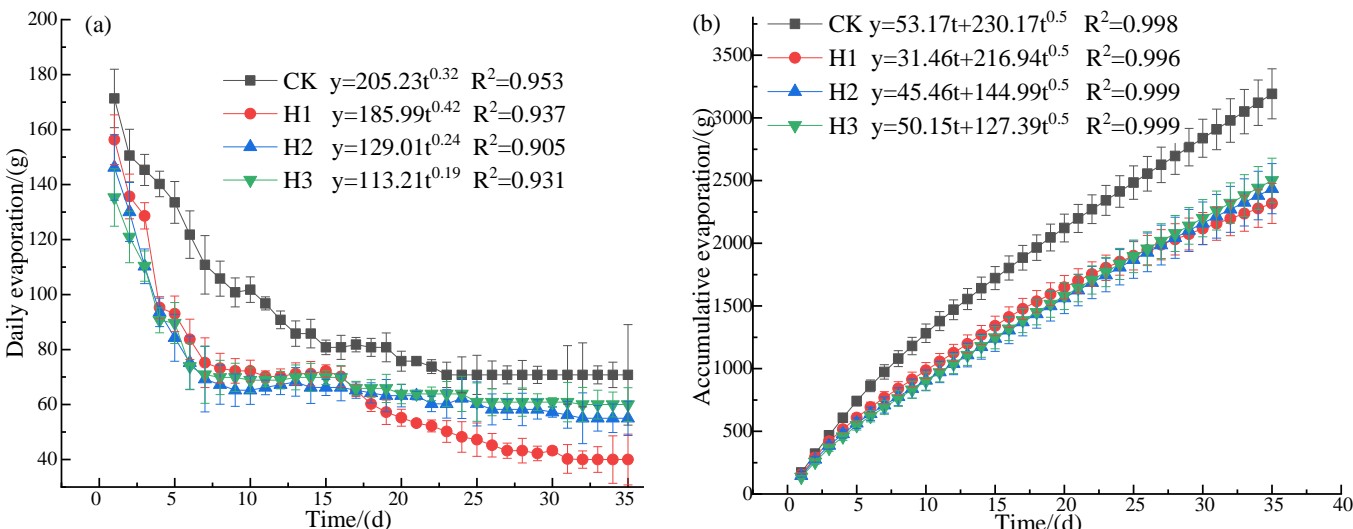

**Figure 5.** Effect of the biochar interlayer on evaporation intensity and cumulative evaporation. (**a**,**b**) represent the changes in evaporation intensity and cumulative evaporation for different treatments, respectively.

The influence of the carbon interlayer on cumulative evaporation is shown in Figure 5b. The cumulative evaporation of the CK soil column was the highest throughout the evaporation period, while that of the H1 soil column in the interlayer treatment was the highest in the first 25 days. As the evaporation process proceeded, the cumulative evaporation of the H1 soil column was gradually reduced compared to that of the H2 and H3 soil columns after 25 days. After 35 days of continuous evaporation, the cumulative evaporation of the CK soil column was 3191.73 g, whereas the values of the cumulative evaporation of H1, H2, and H3 soil columns were 37.75%, 31.05%, and 27.64% lower than that of CK, respectively.

### 3.5. Changes in Water and Salt Content in the Biochar Interlayer

With increasing the compartment depth, the water content of the biochar compartment significantly increased ($p < 0.05$), and as the evaporation went by, the water content of the biochar compartment gradually decreased. After irrigation, the water content of the biochar interlayer in the H3 soil column was 184.80%, higher than that in H1 and H2

soil columns (51.76% and 24.19%, respectively). After 35 days of evaporation, the water content of the biochar interlayer significantly varied between the treatments. The biochar interlayer water content of soil column H3 was 1.41 and 2.26 times that of soil columns H1 and H2, respectively.

The changing trends of the salt and water contents of the biochar compartment were consistent, with a significant difference ($p < 0.05$), showing that H3 > H2 > H1, with the salt content having values of 13.76 g·kg$^{-1}$, 8.83 g·kg$^{-1}$, and 2.31 g·kg$^{-1}$, respectively. With the progress of evaporation, the salt content of the biochar interlayer gradually decreased. After 35 days of evaporation, the salt content of each compartment treated with biochar decreased to varying degrees compared with that before evaporation, 57.19%, 74.79%, and 70.57% of the initial salt content values in H3, H2, and H1, respectively. The relationship between salt adsorption and the water content of the biochar compartment was a power function (Figure 6c).

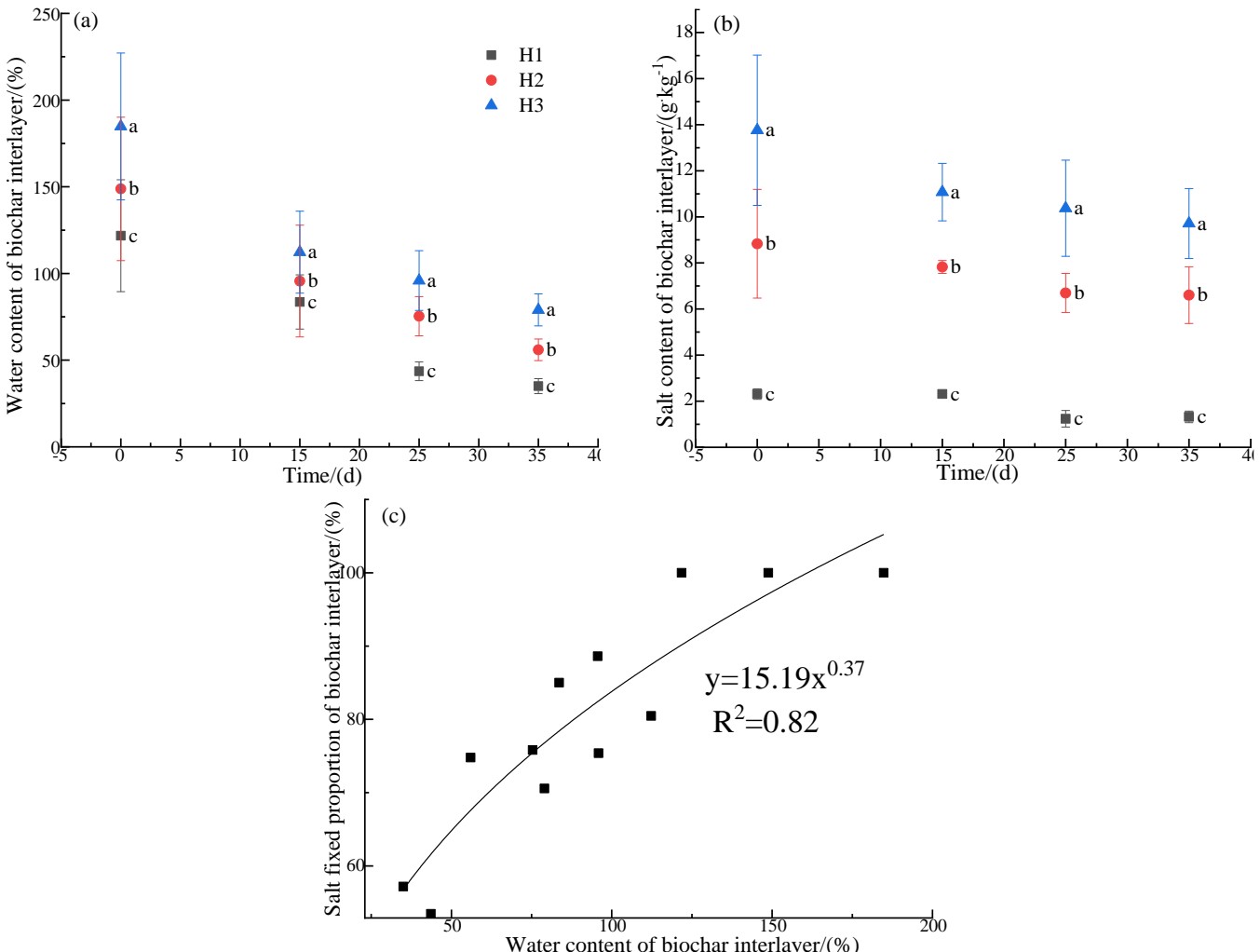

**Figure 6.** Water and salt contents of the biochar interlayer during different evaporation periods; (**a**,**b**) represent the changes in water and salt contents of the biochar interlayer with evaporation time at different burial depths, respectively, and (**c**) represents the fitting curve of the relationship between water and salt contents of the biochar interlayer in different evaporation periods. Lowercase letters in (**a**,**b**) indicate significant differences ($p < 0.05$).

## 4. Discussion

### 4.1. Effect of the Biochar Interlayer on Infiltration Characteristics

In this study, the presence of the biochar interlayer rapidly reduced the cumulative infiltration amount of the H1, H2, and H3 soil columns per unit of time, and the average infiltration rates were 38.46%, 50.00%, and 67.44% lower than that of the CK soil column, respectively. Consistent with the research results of the study conducted by Zhao and others [6], in this study, the barrier formed by biochar provided the soil capillary cut-off, changed the homogeneity of soil texture and soil structure, interrupted the continuity of the soil permeability and water electrical conductivity, and established a discontinuous water transport pathway, which reduced not only the soil infiltration capacity, but also the evaporation capacity [29]. The "pore difference interface" formed by the biochar interlayer and the soil layer caused the difference in water conductivity [30], which decreased the amount of water flowing into the barrier per unit of time, thus reducing the advancing velocity of the wetting front and the rate of soil water infiltration, and consequently, reducing the cumulative infiltration amount per unit of time [31]. In this study, when the infiltrated water entered the interlayer, the cumulative infiltration amount of the H1, H2, and H3 soil columns per unit of time rapidly decreased, 37.60%, 63.02%, and 83.85% lower than that of the CK soil column, respectively. The shallower the interlayer, the less affected the infiltration rate of the soil column, while the deeper the interlayer was buried, the lower the escape rate of pore gas was. A large number of bubbles are stored in the interlayer [32], thereby reducing the water infiltration rate.

When the peak of the wetting front passes through the barrier, the water nonuniformly flows as fingers. Due to the existence of macropores in the biochar interlayer, a difference in water conductivity is caused. The macropore structure in the barrier facilitates the water flow through the preferential pathway, causing the phenomenon of "preferential flow". The water first passes through the barrier into the lower surface of the interface. The air trapped in the barrier increases the air pressure ahead of the wetting front, resulting in a non-uniform flow with a large difference in velocity [33,34]. In addition, due to the large specific surface area and also a large number of pores on the biochar surface, the water adsorption capacity was increased (Figure 3), and thus, the heterogeneity of the wetting front migration was enhanced. During the infiltration process, the soil moisture content increased in deeper layers. When the moisture content of the biochar interlayer reached a certain amount, its hydraulic conductivity tended to be the same as that of the soil layer; the water infiltration rate was stable, but the non-uniformity of the wetting front gradually disappeared.

### 4.2. Effect of the Biochar Barrier on Water and Salt Contents of the Soil Profile

In this study, due to the existence of the biochar interlayer, a water distribution model of "soil carbon from a two-layer soil structure" was formed [35], changing the spatial distribution pattern of water and optimizing the water content of the soil layer above the interlayer (7.79–13.68% higher than that of CK) (Figure 2). This is consistent with the results of the study by Zhang, who used straw deep burial to increase the water content of the upper soil layers [36]. During the process of water flow, the air trapped in the pores of the barrier layer cannot be completely drained or the infiltrated water moves from the surface into the soil, forming entrapped bubbles so that its water content does not reach saturation point. At this time, the conductivity of the biochar barrier layer is lower than that of the upper soil. As a result, the water potential difference between the interlayer and the upper layer of soil cannot be balanced in a short time, and the water movement speed is reduced, which hinders the infiltration process of irrigation water and increases the retention time of irrigation water in the soil layer above the interlayer [32,37]; biochar itself enhances water retention capacity, reducing the water content of the soil layer below the interlayer.

In this study, the cumulative evaporation of the H1, H2, and H3 soil columns decreased by 37.75%, 31.05%, and 27.64%, respectively, compared with that of CK. Zhao et al. [7] believed that in homogeneous soil, the infiltration water leaked before reaching the soil

water-salt diffusion balance, and the salt leaching efficiency per unit volume of water was poor. After the straw interlayer was buried, the air trapped in the large pores in the interlayer formed a barrier layer for water movement, thus extending the storage time of the infiltration water in the soil layer above the interlayer (Figure 2) [31]. The water content of the soil layer above the interlayer increased, and the ion exchange, adsorption, and analysis were also promoted [38] so that the salt could be fully dissolved and the salt leaching efficiency was improved. These findings were consistent with the results of this study. The total salt content of the soil layer above the interlayer in the H1, H2, and H3 soil columns had values of 21.89%, 7.38%, and 26.42% lower than that of CK, respectively, whereas the cumulative salt content of 0–30 cm soil layer had the values of 47.10%, 33.23%, and 26.42% lower than that of CK, respectively.

During the evaporation process, the dehydration process of each soil column was different. The water content of all soil layers in the CK soil column was evenly reduced, and the water content of the soil layers above the interlayer in the H1, H2, and H3 soil columns was also significantly reduced, while the water content below the interlayer remained at a high level since the capillary force effect of the interlayer was extremely weak, and its water conductivity was far lower than that of the CK soil column [32]. The internal water content of the biochar interlayer in the H1 and H2 soil columns decreased (Figure 3), causing the loss of water in most of the pores in the soil column, forming a barrier layer that did not conduct water. When the capillary water in the bottom soil rose to the barrier layer, it could not cross the barrier layer and be transferred [39]. Therefore, the soil water loss above the barrier layer in the H1 and H2 soil columns was relatively high, while the internal water content in the barrier layer of the H3 soil column remained high (Figure 3), and the water content in the 0–30 cm soil layer was also high.

This study shows the prominent effect of the shallow biochar interlayer on the evaluated parameters. After 35 days of continuous evaporation, the cumulative evaporation of the CK soil column was 3191.73 g, while the cumulative evaporation of the H1, H2, and H3 soil columns had values of 37.75%, 31.05%, and 27.64% lower than that of the CK, respectively, and the anti-salt accumulated amounts on the soil surface (0–10 cm) were 53.32%, 38.30%, and 46.07% lower than that of the CK soil column. This is consistent with the results of the study by Yao et al. [40], who pointed out that the shallow straw interlayer had more prominent inhibitory effects on evaporation and accumulation of salt because the water content of the biochar interlayer and the soil layer above the barrier layer was high at the beginning of the evaporation process, and the water loss in the soil layer above the biochar interlayer was rapid. However, since the capillary force effect of the barrier was extremely weak, only the soil above the barrier could be used for evaporation. Therefore, with the deepening of the biochar interlayer, the water loss in the soil layer increased, and the evaporation capacity also gradually increased [38].

### 4.3. Changes in Water and Salt Contents of the Biochar Interlayer

The porous structure and high specific surface area of biochar enable it to fully absorb water and salt in the saline-alkali soil [41–44], which confirms the results of this study. After irrigation, the water content of the biochar interlayer in the H1, H2, and H3 soil columns had values of 184.80%, 148.81%, and 121.77%, respectively, and the values of the salt content were 13.76 g·kg$^{-1}$, 8.83 g·kg$^{-1}$ and 2.31 g·kg$^{-1}$, respectively. The difference in the water and salt contents was caused by the depth of the barriers. The results showed that the biochar interlayer could store water and adsorb salt for a long time. On the one hand, it can play a role in water conservation and salt removal for the soil layer below the barrier (Figures 3 and 4), and on the other hand, the salt leached from the upper layer can be adsorbed onto the biochar interlayer (Figure 6). Even after 35 consecutive days of evaporation, the water content of the biochar compartment in the H1, H2, and H3 soil columns remained at a high level, 79.00%, 56.00%, and 35.02%, respectively. The salt adsorbed onto the compartments after evaporation had contents of 57.19%, 74.79%, and 70.57%, respectively, compared to the initial salt content.

China is the world's largest producer of crop straw, accounting for nearly one-fifth of the global straw resources [45,46]. In terms of comprehensive utilization, it is still at a relatively extensive stage, at which about 30% of the straws are directly burned, which makes it very easy to reduce soil fertility, cause environmental pollution, and produce resource waste [47]. About 70% of the straws are crushed and then buried, crushed and mixed, or covered. However, straw returning in arid and semi-arid areas affected sowing and seed emergence and aggravated the problems caused by diseases, pests, and weeds [48,49]. The effect of deep straw mulching on water conservation and salt accumulation inhibition was stronger than that of surface straw mulching [14], but the high investment costs and short-term efficiency are not conducive to its large-scale promotion and use. Biochar has a high carbon content and stable physical and chemical properties [19], which can stay unchanged in soil for hundreds of years [20]. Using biochar as an interlayer could not only preserve water and control salinity, but also significantly improve the shelf life of the barrier and reduce inputs such as labor and machine, as well as investment costs. Therefore, it can be used as an effective means to improve the saline-alkali land in arid and semi-arid areas.

## 5. Conclusions

In this paper, through the indoor soil column simulation test, the effects of different buried depths of the biochar interlayer on soil water and salt transport during soil water infiltration and evaporation were studied, and the following conclusions were drawn:

(1)   The biochar barrier decreased the soil water infiltration capacity.
(2)   The biochar interlayer optimized the distribution of soil water and salt.
(3)   The biochar interlayer inhibited soil water evaporation.
(4)   The biochar barrier played a role in long-term water storage and salt adsorption.

In summary, the H1 and H2 soil columns could reduce the salt content of the soil layer above the interlayer, but water retention was not overly efficient. The soil above the interlayer in these two soil columns was in a state of "low water and low salt". The inhibition effect of the H3 treatment on soil evaporation and soil salt migration was significantly greater than that of the H1 and H2 treatments, and the soil above the interlayer was in a state of "high water and low salt". The biochar interlayer itself could play a dual role in adsorbing the salt leached from the upper layer and inhibiting the anti-salt migration from the lower layer, which can provide a theoretical basis and reference for water conservation and salt accumulation inhibition, and thus, the improvement of the saline farmland in arid and semi-arid regions.

**Author Contributions:** Conceptualization, Q.X. and H.L.; methodology, Q.X.; software, Q.X.; validation, Q.X., H.L. and M.L.; formal analysis, Q.X.; investigation, Q.X.; resources, Q.X. and H.L.; data curation, Q.X.; writing—original draft preparation, Q.X.; writing—review and editing, Q.X., M.L. and H.L.; supervision, P.L.; project administration, H.L.; funding acquisition, H.L. All authors have read and agreed to the published version of the manuscript.

**Funding:** This research was jointly supported by the National Natural Science Foundation of China (No. 52069026; 51790533; U1803244) and Xinjiang Production and Construction Corps (No. 2022CB002-02).

**Institutional Review Board Statement:** Not applicable.

**Informed Consent Statement:** Not applicable.

**Data Availability Statement:** Not applicable.

**Acknowledgments:** Thank you to all participants for their strong support.

**Conflicts of Interest:** The authors declare no conflict of interest.

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
