# Peer review of "The Presence of the Biochar Interlayer Effectively Inhibits Soil Water Evaporation and Salt Migration to the Soil Surface"

_agriculture, doi:10.3390/agriculture13030638_

Round 1

Reviewer 1 Report

The manuscript entitled ”The Presence of the Biochar Interlayer Effectively Inhibits Soil-Water Evaporation and Salt Migration to the Soil Surface” is a detail investigation. However, some suggestions and justifications are mentioned below for the improvement of the manuscript:

1.      Why didn’t authors choose the soil from salinized land? Is there is any reason of choosing the soil sample from that site?

2.      The authors didn’t mention about the physio-chemical properties of biochar. Please add some details regarding the same and correlate it within the discussion.

3.      Conclusion needs to be rewrite. As authors sum-up by writing the results again and didn’t provide the larger picture of the outcomes. It didn’t seem justified.

4.      Please add references for Wilkes’s method. Soil strength and cumulative soil water evaporation protocols.

5.       The manuscript discussion is needing some improvement as it describes data correlation very well but lack the interpretation in detail with new references. I hope this paper could be useful to improve paper: DOI 10.1038/s41598-023-27638-9

Reviewer 2 Report

Review report of research article "The Presence of the Biochar Interlayer Effectively Inhibits Soil Water Evaporation and Salt Migration to the Soil Surface"

Comments

1. Please check the units. Units is not consistent in literature and in figures.

2. The placement of . very next to units like (cm h-1 ) to avoid plagiarism. Line no 15, 44, 85, 108, 156, 261, 285, 425 and 426.

3. Line no. 118, 119, 138 and 281. Rephrase the sentence.

4. Figure 3, 4 and 6 units not consistent in x and y axis and in title of figures..

5. The % units in figures should be used like this (%)

6. Line no 393. Again units not consistent. 

7. Some reference in reference section missing journal and page numbers. 

Reviewer 3 Report

It is a very good and interesting work with significant findings. My very few comments are included in the attached pdf file.

Author Response

Author's answer:The content about wilkes method is deleted and the physical and chemical properties of soil are supplemented.The physical and chemical properties of soil and biochar are added

Table 1.Physical and chemical properties of soil and biochar

texture

Soil bulk density

/ g·cm-3

Field water holding capacity/%

Conductivity

/ms·cm-1

Organic carbon

/g·kg-1

Total N

/g·kg-1

Total P

/g·kg-1

K

/g·kg-1

Ca

/g·kg-1

Mg

/g·kg-1

Silty loam

1.40

26.46

2.56

4.63

0.39

0.75

12.35

6.28

4.51

Biochar

0.51

0.36

521.69

25.72

11.02

20.58

19.63

4.26

Reviewer 4 Report

Review of the manuscript: The Presence of the Biochar Interlayer Effectively Inhibits Soil Water Evaporation and Salt Migration to the Soil Surface

General comments: The manuscript is well written in English and reports the effects of the biochar interlayer on soil water infiltration and soil water evaporation and unravel its regulation mechanism of soil water and salt movement to provide a theoretical basis and reference for water conservation and salt removal in the saline farmland in arid and semi-arid areas. The introduction, methods, results are well documented and the manuscript can be published.

Comments to authors:

METHODS:

Please describe the soil type used in the experiment

It would be nice to include a photo of the experimental device (soil column)
